# Lower rates of ART initiation and decreased retention among ART-naïve patients who consume alcohol enrolling in HIV care and treatment programs in Kenya and Uganda

**Ioannis Patsis[1], Suzanne Goodrich[2], Constantin T. Yiannoutsos[3]\*, Steven A. Brown[4], Beverly S. Musick[4], Lameck Diero[5], Jayne L. Kulzer[6], Mwembesa Bosco Bwana[7], Patrick Oyaro[8], Kara K. Wools-Kaloustian[2]**

1 Department of Hygiene and Epidemiology, School of Medicine, National and Kapodistrian University of Athens, Athens, Greece, 2 Division of Infectious Diseases, Department of Medicine, Indiana University School of Medicine, Indianapolis, Indiana, United States of America, 3 Department of Biostatistics, Indiana University Fairbanks School of Public Health, Indianapolis, Indiana, United States of America, 4 Department of Biostatistics, Indiana University School of Medicine, Indianapolis, Indiana, United States of America, 5 School of Medicine, College of Health Sciences, Moi University, Eldoret, Kenya, 6 Department of Medicine, University of California San Francisco, San Francisco, California, United States of America, 7 Department of Medicine, Mbarara University of Science and Technology, Mbarara, Uganda, 8 Centre for Microbiology Research, Kenya Medical Research (KEMRI), Nairobi, Kenya

\* cyiannou@iu.edu

**Data Availability Statement:** Regarding data sharing, complete data for this study cannot be publicly shared because of legal and ethical

## Abstract

### Objectives

Almost 13 million people are estimated to be on antiretroviral therapy in Eastern and Southern Africa, and their disease course and program effectiveness could be significantly affected by the concurrent use of alcohol. Screening for alcohol use may be important to assess the prevalence of alcohol consumption and its impact on patient and programmatic outcomes.

### Methods

As part of this observational study, data on patient characteristics and alcohol consumption were collected on a cohort of 765 adult patients enrolling in HIV care in East Africa. Alcohol consumption was assessed with the AUDIT questionnaire at enrollment. Subjects were classified as consuming any alcohol (AUDIT score >0), hazardous drinkers (AUDIT score $\geq$8) and hyper drinkers (AUDIT score $\geq$16). The effects of alcohol consumption on retention in care, death and delays in antiretroviral therapy (ART) initiation were assessed through competing risk (Fine & Gray) models.

### Results

Of all study participants, 41.6% consumed alcohol, 26.7% were classified as hazardous drinkers, and 16.0% as hyper drinkers. Depending on alcohol consumption classification, men were 3–4 times more likely to consume alcohol compared to women. Hazardous drinkers (median age 32.8 years) and hyper drinkers (32.7 years) were slightly older compared to

restrictions. The principles of collaboration under which IeDEA was founded and the regulatory requirements of the different countries' IRBs and other legislative and regulatory bodies, require the submission and approval of a project concept sheet by investigators, both within and outside of IeDEA, which has to be approved by the individual sites and the IeDEA Regional Executive Committee. Proposals to individual regions are governed by similar processes (see https://www.ccasanet.org/collaborate/ for helpful documents and processes governing the Central, South America and the Caribbean Network, one of the seven IeDEA regions as well the concept proposal form for the East Africa IeDEA region, from where this manuscript originated). The website (www.iedea-ea.org) includes all information and provides access to forms that researchers in the community can use to request data from IeDEA East Africa. Requests can be directed to our project coordinator, Ms. Yee Yee Kuhn (ykuhn@iu.edu).

**Funding:** CTY, KWK, LD, JK, MB and BSM were funded by grant number AI069911 provided by the National Institutes of Allergy and Infectious Diseases SG work was funded by a supplement to this grant provided by the National Institute of Drug Abuse. IP was funded by a graduate scholarship offered by the National and Kapodistrian University of Athens Greece.

**Competing interests:** The authors have declared that no competing interests exist.

non-hazardous drinkers (30.7 years) and non-hyper drinkers (30.8 years), (p-values = 0.014 and 0.053 respectively). Median CD4 at enrollment was 330 cells/µl and 16% were classified World Health Organization (WHO) stage 3 or 4. There was no association between alcohol consumption and CD4 count or WHO stage at enrollment. Alcohol consumption was associated with significantly lower probability of ART initiation (adjusted sub-distribution hazard ratio aSHR = 0.77 between alcohol consumers versus non-consumers; p-value = 0.008), and higher patient non-retention in care (aSHR = 1.77, p-value = 0.023).

## Discussion

Alcohol consumption is associated with significant delays in ART initiation and reduced retention in care for patients enrolling in HIV care and treatment programs in East Africa. Consequently, interventions that target alcohol consumption may have a significant impact on the HIV care cascade.

## Introduction

In 2017, there were 19.6 million [17.5 million– 22.0 million] people with HIV (PWH) in Eastern and Southern Africa [1]. As the number of patients accessing HIV care has rapidly increased in the past ten years, nearly 12.9 million people [11.4 million- 13.4 million] in Eastern and Southern Africa are now on antiretroviral therapy (ART) [1]. However, retention in care remains a significant challenge [2]. For example, in East Africa only about 69% of patients initiating ART remain in care at the clinic of their initial enrollment after two years[3]. Patients who are disengaged from care have significantly higher mortality [2, 4, 5] and HIV transmission rates compared to those who remain engaged in care [6, 7]. Consequently, the investigation of factors that impact patterns of ART initiation and retention in care for patients enrolling in these programs is of particular significance.

Though the association between alcohol consumption and poor health outcomes in HIV infection is well established [8–10], the impact of alcohol consumption on patients' initiation of ART and retention in care has yet to be thoroughly examined. Nevertheless, there is some recent evidence that drinking is associated with poor retention in care, in both resource-replete and resource-limited settings [11, 12]. Heavy drinking and frequent binge drinking are associated with worse retention in HIV care in patients observed at seven U.S. HIV clinical sites [11], and findings from a systematic review suggest that low-income and middle-income countries (as classified by the World Bank) face issues similar to those of high-income countries [12]. Despite that, only a few evidence-based interventions specifically target problematic alcohol consumption [12]. In addition, measurement of alcohol exposure and retention measures varies among studies so that generalizability of their findings is an issue. Moreover, adjusting for all possible confounders in such studies is challenging, and causality cannot be inferred [11]. In Africa, 42.6% of the population is estimated to consume alcohol [13]. Thus, the impact of alcohol on the course of HIV infection and the effectiveness of care and treatment programs, which rely on rapid initiation of patients on ART and patient retention into care for their success, may be significantly affected by ambient alcohol consumption patterns, particularly in resource-limited areas with high HIV prevalence like sub-Saharan Africa.

The purpose of this observational study was to determine the prevalence of alcohol consumption in ART-naïve patients initiating HIV care and to assess whether alcohol consumption is associated with time to ART initiation, mortality, and patient retention in care.

## Methods

### Study design

This prospective observational study was approved by the Indiana University Institutional Review Board and the ethical bodies affiliated with each participating site: The Academic Model Providing Access to Healthcare (AMPATH): Moi University College of Health Sciences and MOI Teaching and Referral Hospital's Institutional Research and Ethics Committee; Family AIDS Care and Education Services (FACES): Kenya Medical Research Institute/National Ethics Review Committee; Mbarara Immune Suppression Syndrome (ISS) Clinic: Mbarara University of Science & Technology Institutional Review Committee. Participant written informed consent was obtained at the time of enrollment into study.

### Study setting and standard of care

The study took place in five clinics within the East Africa International epidemiology Databases to Evaluate AIDS (EA-IeDEA) consortium. EA-IeDEA is one of seven regional consortia supported by the National Institutes of Health to consolidate, curate and analyze HIV care and treatment data in order to evaluate the outcomes of people living with HIV/AIDS [14]. Study sites included the FACES clinics at Lumumba sub-County Hospital, (Kisumu Kenya) and Suba sub-County Hospital (Sindo, Kenya); two AMPATH clinics based at Moi Teaching and Referral Hospital (Eldoret, Kenya); and the Mbarara ISS Clinic (Mbarara, Uganda). The sites in Eldoret and Kisumu are primarily urban, while Sindo is rural and Mbarara is semi-urban. Each clinic provides comprehensive HIV care including CD4 and HIV viral load testing, provision of ART, diagnosis and treatment of opportunistic infections, as well as management of common non-communicable diseases.

Individuals enrolling in care have blood drawn for a CD4 count and undergo World Health Organization (WHO) disease staging by a clinician [15]. Patients started on trimethoprim-sulfa prophylaxis, if no contraindication existed. Eligibility criteria for ART was assessed as per WHO guidelines during the study period, hence individuals were eligible for ART if they had a CD4 count below 350 cells/μl and/or a WHO stage of >2 [16].

Patients not meeting ART eligibility criteria returned to clinic at a frequency determined by the clinic, but usually at a maximum of every six months, for care and continued monitoring of their CD4 count for ART eligibility. For patients initiating ART, CD4 counts were repeated at 6–12 months depending on the clinics testing strategy. Routine viral load testing was not always available during the study period.

### Study population

Participants were recruited from January 25, 2013 to June 25, 2014. Participants met eligibility criteria for inclusion in this study if they were at least 18 years of age, newly enrolling in HIV-care at one of the five participating clinics, and ART-naïve. Potential participants were excluded if they did not meet any of the above criteria or were unable or unwilling to provide consent for the study. All participants were educated about the study and provided written consent prior to enrollment.

The Alcohol Use Disorder Identification Test (AUDIT) questionnaire, which has been validated in a number of primary care medical settings, was used by a trained research assistant to collect patient's alcohol use data over the past one year [17–20]. The AUDIT version utilized is comprised of 10 questions covering three domains: hazardous alcohol use, dependence symptoms, and harmful alcohol use. The total AUDIT score is calculated by summing the values from each question (range 0–4). RAs attempted to minimize under-reporting of a participant's alcohol use by asking in-depth questions about the types of alcohol consumed, the volume

(facilitated by showing participants a large collection of glasses and bottles in varying shapes and sizes) and frequency of consumption. Using a spreadsheet with the known alcohol content of commercially available drinks and estimates of the alcohol content of local (non-commercial) brews [21], calculations were made to determine the number of standard drinks (one drink defined as 10 grams of alcohol) each participant consumed.

Additional information was collected as part of routine care including demographics (e.g., age, gender, civil status, HIV disclosure), and clinical data (e.g., CD4 count and WHO disease stage at enrollment). Locator information including addresses and telephone numbers were obtained as part of clinic enrollment and then verified and updated as needed by study personnel. Participants who failed to return for a scheduled visit for more than two months were traced by trained outreach personnel. The first attempt was made by telephone, and, if unsuccessful, community tracing was attempted. Once the participant or a reliable close informant was found, vital status and engagement in care were documented.

## Statistical methods

**Definition of the outcomes of interest.** The three outcomes of interest were time from enrollment in care to ART initiation, mortality, and retention in care. Retention in care was measured by rates of patients actually disengaged from care as ascertained by tracing patients in the community who failed to return to clinic within two months from their last clinic visit to ensure they had not died or were receiving care elsewhere [22, 23]. In this manner, we were able to estimate patient retention in care from the perspective of the patient rather than that of the program. On the basis of tracing and community outreach, participants were classified as silent transfers (undocumented transfer to another care facility), or as having relocated, deceased, disengaged, or missing (untraceable). Within these analyses, censoring occurred at the date patients were traced or, if not located, at the last clinic visit for patients who were determined to be silent transfers and participants who had relocated, because these outcomes were not considered adverse outcomes from a programmatic perspective and were classified as being retained in care. By contrast, patients who were untraceable (missing) or who were found to have disengaged from care were considered as having adverse programmatic outcomes and so were classified as not having been retained in care.

**Predictors used in the analyses.** Utilizing WHO criteria binary categories based on AUDIT scores were developed: non-alcohol consumers (AUDIT = 0) versus alcohol consumers (AUDIT > 0); people engaged in hazardous drinking (AUDIT ≥ 8) versus not (AUDIT < 8); and people engaged in high alcohol consumption (combined categories of harmful drinking and risk for dependence; AUDIT ≥16) versus not (AUDIT <16) [13].

The covariates of interest included gender, age (18–24, 25–34, 35–44, 45+), civil status (legally married versus not), CD4 cell count at enrollment (<50, 50–99, 100–199, 200–349, 350–499, 500+), WHO stage (1/ 2 or 3/ 4) and HIV status disclosure (whether the participant has disclosed HIV carrier status to anyone) at the time of enrollment. The five participating clinics were grouped under their parent programs: FACES, AMPATH and Mbarara. Missing values for CD4 cell count, WHO stage, civil status, and HIV disclosure multiple imputation was utilized using iterative chained equations (ICE) under the assumption that the missing values were missing at random (MAR) [24].

**Statistical modeling.** Descriptive statistics were used to calculate the prevalence of alcohol consumption. The association between age, CD4 count at enrollment and alcohol consumption was examined by the Mann-Whitney test. The Pearson chi-square test was used to examine the possible association between categorical variables like gender, WHO stage, age and the AUDIT alcohol consumption categories.

To examine the impact of alcohol use on delays in ART initiation, retention in care and mortality, survival analysis techniques were used. Study participants were at risk of experiencing more than one outcome. However, observation of one outcome such as death or non-retention, prior to another, such as, for example, ART initiation, would preclude observing the other outcome. As such, all events other than the event of interest were considered competing risks [25]. To assess the impact of alcohol consumption on each event of interest, in the presence of all competing events and other explanatory factors, we used the method of Fine and Gray in the analysis [25]. We performed three separate analyses, each investigating the possible effect of different levels of alcohol consumption (i.e., any alcohol use, hazardous and higher or harmful alcohol consumption, the exposure) on the likelihood (sub-distribution hazard) of each of the three events of interest (ART initiation, death, and non-retention in care). In all cases we adjusted for all relevant patient-level characteristics. With regard to the event of ART initiation, the competing risks were non-retention in care and death, whereas for mortality, the corresponding competing risks were non-retention and ART initiation. The competing risk for the event of non-retention was death.

Time-to-event analyses were conducted under the assumption of independent censoring. This means that the failure rate for subjects within a subgroup (e.g., people engaged in hazardous alcohol consumption) who had relocated or silently transferred is assumed to be equal to the failure rate for subjects in that subgroup who remained on observation and had not experienced any event until that time-point [26]. Associations with corresponding p-values <5% were designated as statistically significant, and as possibly significant if between 5% and 10%. Data analyses were conducted using the statistical program Stata version 13 (StataCorp, College Station, Texas).

## Results

### Participant characteristics

A total of 765 patients with median age of 31.2 years at enrollment participated in the study (Table 1); more than half were women (61%). At enrollment, the median CD4 count was 330 cells/µl (IQR: 137–513) and 84% (n = 591) of study participants with a known WHO stage had a WHO stage 1 or 2 disease. About two thirds of the participants with a known civil status were legally married (66%, n = 361) and just over half of those with a known disclosure status had disclosed their HIV status (56%, n = 269). As classified by AUDIT scores, 41.6% of study participants consumed alcohol, with 26.7% scoring at least in the hazardous range and 16.0% in the harmful drinking range or being at risk for dependency.

Men were more likely than women to consume alcohol regardless of the categorization of alcohol consumption (Table 1). In fact, the odds of belonging to the higher alcohol consumption category were between 3-fold and 4-fold higher for men compared to women. Older participants were more likely to belong to higher alcohol consumption categories when compared to younger participants. However, no age-by-gender interaction was identified. There was no association between enrollment CD4 count, WHO stage, or HIV disclosure status and alcohol consumption (Table 1).

Initially, 25% (n = 190) of participants were identified as lost to program. After tracing, 38% were found to be silent (unreported) transfers, 29% disengaged from care, 7% deceased, 12% relocated and 13% untraceable (missing). During the study period, 28 (3.7%) participants died.

### Competing-risk analysis of the events of interest

Consumption of alcohol was strongly associated with a higher (sub-distribution) hazard of not being retained in care in a competing-risk analysis with death as the competing event.

**Table 1. Characteristics of the study cohort.**

| | Overall (n:765) | no alcohol consumption | alcohol consumption[1] | non-hazardous alcohol consumption | Hazardous alcohol consumption[2] | Non-harmful drinking | Harmful drinking[3] |
|---|---|---|---|---|---|---|---|
| **Overall (n:765)** | | 447 (58.4) | 318 (41.6) | 561(73.3) | 204 (26.7) | 643 (84.1) | 122 (16.0) |
| **IeDEA program p-value*** | | <0.001 | | 0.062 | | <0.001 | |
| AMPATH | 264 (34.5) | 166 (37.1) | 98 (30.8) | 184 (32.8) | 80 (39.2) | 209 (32.5) | 55 (45.1) |
| Mbarara | 264 (34.5) | 130 (29.1) | 134 (42.1) | 207 (36.9) | 57 (27.9) | 245 (38.1) | 19 (15.6) |
| FACES | 237 (31.0) | 151 (33.8) | 86 (27.0) | 170 (30.3) | 67 (32.8) | 189 (23.4) | 48 (39.3) |
| **Total** | 765 | 447 | 318 | 561 | 204 | 643 | 122 |
| **Gender [N (%)] p-value*** | | <0.001 | | <0.001 | | <0.001 | |
| Male | 295 (38.6) | 120 (26.9) | 175 (55.0) | 164 (29.2) | 131 (64.2) | 213 (33.1) | 82 (67.2) |
| Female | 470 (61.4) | 327 (73.2) | 143 (45.0) | 397 (70.8) | 73 (35.8) | 430 (66.9) | 40 (32.8) |
| **Total** | 765 | 447 | 318 | 561 | 204 | 643 | 122 |
| **Age [median (IQR)] p-value**** | | NS | | 0.014 | | 0.053 | |
| | 31.2 (26.1, 39.5) | 30.8 (25.8, 39.8) | 32.0 (26.9, 39.4) | 30.7 (25.8, 38.9) | 32.8 (27.6, 40.5) | 30.8 (25.8, 39.4) | 32.7 (27.8, 40.7) |
| **Age (categorical) [N (%)] p-value*** | | NS | | NS | | NS | |
| 18–24 | 147 (19.2) | 97 (21.7) | 50 (15.7) | 120 (21.4) | 27 (13.2) | 134 (20.8) | 13 (10.7) |
| 25–34 | 338 (44.2) | 186 (41.6) | 152 (47.8) | 243 (43.3) | 95 (46.6) | 279 (43.4) | 59 (48.4) |
| 35–44 | 182 (23.8) | 107 (23.9) | 75 (23.6) | 128 (22.8) | 54 (26.5) | 147 (22.9) | 35 (28.7) |
| >44 | 98 (12.8) | 57 (12.8) | 41 (12.9) | 70 (12.5) | 28 (13.7) | 83 (12.9) | 15 (12.3) |
| **WHO stage at enrollment [N (%)] p-value*** | | NS | | NS | | NS | |
| 1–2 | 591 (84.0) | 341 (84.2) | 250 (83.6) | 437 (85.2) | 154 (80.6) | 500 (84.6) | 91 (80.5) |
| 3–4 | 113 (16.1) | 64 (15.8) | 49 (16.4) | 76 (14.8) | 37 (19.4) | 91 (15.4) | 22 (19.5) |
| **Total (non-missing)** | 704 | 405 | 299 | 513 | 191 | 591 | 113 |
| Missing | 61 (8.0) | 42 (9.4) | 19 (6.0) | 48 (8.6) | 13 (6.4) | 52 (8.1) | 9 (7.4) |
| **CD4 at enrollment [median (IQR)] p-value**** | | NS | | NS | | NS | |
| | 330 (137, 513) | 316 (148, 534) | 342 (127, 487) | 325 (150, 525) | 342 (115, 479) | 329 (141, 515.5) | 330 (124, 469) |
| Missing [N (%)] | 197 (25.8) | 130 (29.1) | 67 (21.1) | 142 (25.3) | 55 (27.0) | 159 (24.7) | 38 (31.1) |
| **CD4 count (categorical) [N (%)]** | | | | | | | |
| [0–49] | 67 (11.8) | 37 (11.7) | 30 (12.0) | 47 (11.2) | 20 (13.4) | 56 (11.6) | 11 (13.1) |
| [50–99] | 45 (7.9) | 23 (7.3) | 22 (8.8) | 32 (7.6) | 13 (8.7) | 38 (7.9) | 7 (8.3) |
| [100–249] | 119 (21.0) | 73 (23.0) | 46 (18.3) | 92 (22.0) | 27 (18.1) | 102 (21.1) | 17 (20.2) |
| [250–349] | 70 (12.3) | 39 (12.3) | 31 (12.4) | 53 (12.7) | 17 (11.4) | 59 (12.2) | 11 (13.1) |
| [350–499] | 117 (20.6) | 57 (18.0) | 60 (23.9) | 79 (18.9) | 38 (22.5) | 97 (20.0) | 20 (23.8) |
| ≥500 | 150 (26.4) | 88 (27.8) | 62 (24.7) | 116 (27.7) | 34 (22.8) | 132 (27.3) | 18 (21.4) |
| **Total** | 568 | 317 | 251 | 419 | 149 | 484 | 84 |
| **Civil status [N (%)] p-value*** | | NS | | NS | | NS | |
| Not married | 189 (34.4) | 101 (32.8) | 88 (36.4) | 139 (34.3) | 50 (34.5) | 165 (35.2) | 24 (29.6) |
| Married | 361 (65.6) | 207 (67.2) | 154 (63.6) | 266 (65.7) | 95 (65.5) | 304 (64.8) | 57 (70.4) |
| **Total (non-missing)** | 550 | 308 | 242 | 405 | 145 | 469 | 81 |
| Missing | 215 (28.1) | 139 (31.1) | 76 (23.9) | 156 (27.8) | 59 (28.9) | 174 (27.1) | 41 (33.6) |
| **HIV status disclosure [N (%)] p-value*** | | NS | | NS | | NS | |
| Not disclosed | 208 (43.6) | 135 (44.9) | 73 (41.5) | 148 (43.8) | 60 (43.2) | 165 (43.4) | 43 (44.3) |

(*Continued*)

**Table 1.** (Continued)

| | Overall (n:765) | no alcohol consumption | alcohol consumption[1] | non-hazardous alcohol consumption | Hazardous alcohol consumption[2] | Non-harmful drinking | Harmful drinking[3] |
|---|---|---|---|---|---|---|---|
| Disclosed | 269 (56.4) | 166 (55.2) | 103 (58.5) | 190 (56.2) | 79 (56.8) | 215 (56.6) | 54 (55.7) |
| **Total (non-missing)** | 477 | 301 | 176 | 338 | 139 | 380 | 97 |
| Missing | 288 (37.6) | 146 (32.7) | 142 (44.7) | 223 (39.8) | 65 (31.9) | 263 (40.9) | 25 (20.5) |

[1]AUDIT score of >0
[2]AUDIT score of ≥8
[3]AUDIT score of ≥16
NS: non-significant
*Chi-square test
**Mann-Whitney test.

Adjusted for all relevant factors, patients who consumed any amount of alcohol had a 77% higher hazard of being non-retained in care (adjusted sub-distribution hazard rate–aSHR–1.77, p-value = 0.023) compared to those not consuming alcohol. Given the low rates of non-retention (10.6%, Table 2), this is approximately equal 77% higher rate of non-retention. Other alcohol exposure classifications (i.e., hazardous versus non-hazardous and harmful versus non-harmful alcohol consumption) were not associated with patient retention (Table 3).

Consumption of alcohol was also associated with a reduced likelihood (sub-distribution hazard) of ART initiation. In the adjusted analysis, patients who consumed any alcohol were approximately 25% less likely to initiate ART as compared to persons who do not consume alcohol (aSHR: 0.77; p-value = 0.008). No association was observed between the other alcohol consumption classifications and the likelihood of ART initiation (Table 3).

Alcohol consumption, regardless of classification, was not associated with an increased hazard of mortality (Table 3).

## Discussion

This study found a strong association between patient's alcohol consumption in the past one year and retention in HIV care. Association between higher levels of alcohol consumption and

**Table 2. Events used in the survival analyses.**

| Main event | Competing event(s) | Alcohol consumption | | | Hazardous drinking | | | Harmful drinking | | |
|---|---|---|---|---|---|---|---|---|---|---|
| | | No | Yes | Total | No | Yes | Total | No | Yes | Total |
| **Total** | | 447 | 318 | 765 | 561 | 204 | 765 | 643 | 122 | 765 |
| *Non-retention [N(%)]* | | 41 (9.2) | 40 (12.6) | 81 (10.6) | 55 (9.8) | 26 (12.8) | 81 (10.6) | 63 (9.8) | 18 (14.8) | 81 (10.6) |
| | Death [N(%)] | 14 (3.1) | 14 (4.4) | 28 (3.7) | 16 (2.9) | 12 (5.9) | 28 (3.7) | 22 (3.4) | 6 (4.9) | 28 (3.7) |
| | Censored [N(%)] | 392 (87.7) | 264 (83.0) | 656 (85.8) | 490 (87.3) | 166 (81.4) | 656 (85.8) | 558 (86.8) | 98 (80.3) | 656 (85.8) |
| *Death [N(%)]* | | 14 (3.1) | 14 (4.4) | 28 (3.7) | 16 (2.9) | 12 (5.9) | 28 (3.7) | 22 (3.4) | 6 (4.9) | 28 (3.7) |
| | Non-retention [N(%)] | 41 (9.2) | 40 (12.6) | 81 (10.6) | 55 (9.8) | 26 (12.8) | 81 (10.6) | 63 (9.8) | 18 (14.8) | 81 (10.6) |
| | Censored [N(%)] | 392 (87.7) | 264 (83.0) | 656 (85.8) | 490 (87.3) | 166 (81.4) | 656 (85.8) | 558 (86.8) | 98 (80.3) | 656 (85.8) |
| *ART initiation [N(%)]* | | 313 (70.0) | 208 (65.4) | 521 (68.1) | 386 (68.8) | 135 (66.2) | 521 (68.1) | 441 (68.6) | 80 (65.6) | 521 (68.1) |
| | Non-retention [N(%)] | 21 (4.7) | 26 (8.2) | 47 (6.1) | 29 (5.2) | 18 (8.8) | 47 (6.1) | 34 (5.3) | 13 (10.7) | 47 (6.1) |
| | Death [N(%)] | 5 (1.1) | 3 (0.9) | 8 (1.1) | 5 (0.9) | 3 (1.5) | 8 (1.1) | 6 (0.9) | 2 (1.6) | 8 (1.1) |
| | Censored [N(%)] | 108 (24.2) | 81 (25.5) | 189 (24.7) | 141 (25.1) | 48 (23.5) | 189 (24.7) | 162 (25.2) | 27 (22.1) | 189 (24.7) |

**Table 3. Results of the Fine & Gray competing events survival models, adjusting for age, gender, CD4 count and WHO stage at enrollment, disclosure of HIV status, marital status and site of enrollment; aSHR: adjusted sub-distribution hazard.**

| Main event | Competing risk | Any alcohol use | | Hazardous alcohol consumption | | Harmful alcohol consumption | |
|---|---|---|---|---|---|---|---|
| | | aSHR [95% CI] | (p-value) | aSHR [95% CI] | (p-value) | aSHR [95% CI] | (p-value) |
| *Non-retention* | Death | 1.766 [1.083, 2.879] | 0.023 | 1.380 [0.737, 2.389] | 0.250 | 1.480 [0.817, 2.681] | 0.195 |
| *Death* | Non-retention | 1.164 [0.580, 2.334] | 0.669 | 1.736 [0.799, 3.773] | 0.163 | 1.245 [0.464, 3.345] | 0.663 |
| *ART initiation* | Non-retention Death | 0.770 [0.636, 0.933] | 0.008 | 0.841 [0.680, 1.040] | 0.111 | 0.814 [0.623, 1.064] | 0.132 |

patient retention in care was less clear, suggesting that it may be the consumption of alcohol *per se* which may be associated with patient retention rather than the amount of consumed alcohol. These results are largely consistent with a large U.S.-based study [11]. However, that study, which included more men (82%) who were identified as heavy drinkers (25% vs. 16% in our study), found a stronger association (adjusted OR 0.78) between heavy alcohol consumption and non-retention, something that our study failed to establish. A systematic review also identified the adverse impact of alcohol use on the HIV care cascade (diagnosis, linkage to care and retention) but only 4 out of 53 studies in that review specifically focused on retention in care and only one was set in a low-income country [12].

Our analysis showed that patients who consumed alcohol had a lower likelihood of initiating ART. In a systematic review [12] three out of six studies looking specifically at ART initiation had similar findings, with the others failing to show a significant association between alcohol consumption and delay in ART initiation. Given that alcohol consumption did not have an effect on ART eligibility in our data (analysis not shown), lower rates of ART initiation among patients that consume alcohol may have occurred because these patients may not have met assessment criteria suggesting readiness to start ART and to maintain adherence to the ART regimen. As a result, providers may have been reluctant to initiate patients who consume alcohol on ART, thus delaying the start of therapy. Interestingly, a study from Uganda [27] showed a majority of patients who were previously consuming alcohol and started on ART subsequently abstained from alcohol over three years of follow-up. While providers may perceive that delay in ART initiation is advantageous, ART start may actually improve abstinence to alcohol and allow for the public health benefit of decreasing patients' viral load.

Alcohol consumption was not associated with higher mortality in our cohort. Mortality was generally a rare event in our study given the short follow-up time (less than one year for most participants) and the fact that the majority of our patients were relatively healthy at enrollment as determined by CD4 count and WHO stage (Table 1). Consequently, with less than 4% of study subjects dying during the study, our study was not powered to detect any potential effects of alcohol consumption on mortality.

We found no association between alcohol consumption and HIV disease severity in patients enrolling in care based on WHO stage and CD4 cell count. These results are consistent with prior studies in which alcohol dependence was associated with lower CD4 counts, but moderate alcohol consumption was not [28–31]. Moreover, given the small proportion of high alcohol consumption in our study, such an effect might be difficult to detect.

Our inability to establish a clear association between the highest levels of alcohol consumption and adverse patient and programmatic outcomes may be due to inaccurate patient reporting despite the significant effort made by our study staff to determine the actual number of standardized drinks consumed. It is possible that some participants attempted to offer more socially acceptable descriptions of their alcohol consumption patterns, which may have resulted in an underestimation of the number of heavy drinkers. In addition, the alcohol content of domestically produced brews varies and thus is difficult to assess. Combined, these

limitations in our data collection may account for finding an association of alcohol consumption with decreased retention in care and delays in ART initiation, but not confirming the same findings among those at the highest alcohol consumption classifications.

## Conclusion

Our results show that alcohol consumption, irrespective of amount, is strongly associated both with lower rates of ART initiation and lower patient retention in care (respectively the second and the critical third component of the WHO's 90-90-90 target [1]). Consequently, its evaluation appears to be a significant target for intervention in HIV care. With a high prevalence of alcohol consumption in sub-Saharan Africa, alcohol use is a factor that programs should consider addressing as they implement and employ universal testing and treatment for all people living with HIV [16].

## Author Contributions

**Conceptualization:** Suzanne Goodrich, Beverly S. Musick, Kara K. Wools-Kaloustian.

**Data curation:** Steven A. Brown, Beverly S. Musick.

**Formal analysis:** Ioannis Patsis.

**Funding acquisition:** Constantin T. Yiannoutsos, Kara K. Wools-Kaloustian.

**Investigation:** Suzanne Goodrich, Lameck Diero, Jayne L. Kulzer.

**Methodology:** Ioannis Patsis, Constantin T. Yiannoutsos, Beverly S. Musick.

**Project administration:** Suzanne Goodrich.

**Resources:** Lameck Diero, Jayne L. Kulzer, Mwembesa Bosco Bwana, Patrick Oyaro, Kara K. Wools-Kaloustian.

**Supervision:** Suzanne Goodrich, Constantin T. Yiannoutsos, Lameck Diero, Jayne L. Kulzer, Mwembesa Bosco Bwana, Patrick Oyaro, Kara K. Wools-Kaloustian.

**Validation:** Suzanne Goodrich, Jayne L. Kulzer, Mwembesa Bosco Bwana, Kara K. Wools-Kaloustian.

**Writing – original draft:** Ioannis Patsis, Suzanne Goodrich, Constantin T. Yiannoutsos, Beverly S. Musick, Kara K. Wools-Kaloustian.

**Writing – review & editing:** Ioannis Patsis, Suzanne Goodrich, Constantin T. Yiannoutsos, Steven A. Brown, Beverly S. Musick, Lameck Diero, Jayne L. Kulzer, Mwembesa Bosco Bwana, Kara K. Wools-Kaloustian.

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
