## [Decision Letter · Decision Letter 0]

18 Dec 2019

PONE-D-19-24950

Prevalence and Impact of Alcohol Use in Patients Enrolling in HIV Care in East Africa

PLOS ONE

Dear Dr. Yiannoutsos,

Thank you for submitting your manuscript to PLOS ONE. After careful consideration, we feel that it has merit but does not fully meet PLOS ONE’s publication criteria as it currently stands. Therefore, we invite you to submit a revised version of the manuscript that addresses the points raised during the review process.

We would appreciate receiving your revised manuscript by Feb 01 2020 11:59PM. To enhance the reproducibility of your results, we recommend that if applicable you deposit your laboratory protocols in protocols.io, where a protocol can be assigned its own identifier (DOI) such that it can be cited independently in the future. For instructions see: http://journals.plos.org/plosone/s/submission-guidelines#loc-laboratory-protocols

We look forward to receiving your revised manuscript.

Kind regards,

Joel Msafiri Francis, MD, MS, PhD

Academic Editor

PLOS ONE

Journal Requirements:

2. Thank you for stating the following in your Competing Interests section: "None".

a. Please state any Competing Interests. If you have no competing interests, please state "The authors have declared that no competing interests exist.", as detailed online in our guide for authors at http://journals.plos.org/plosone/s/submit-now

5. Please amend your list of authors on the manuscript to ensure that each author is linked to an affiliation. Authors’ affiliations should reflect the institution where the work was done (if authors moved subsequently, you can also list the new affiliation stating “current affiliation:….” as necessary).

Reviewers' comments:

Reviewer's Responses to Questions

**Comments to the Author**

1. Is the manuscript technically sound, and do the data support the conclusions?

Reviewer #1: Partly

Reviewer #2: Partly

2. Has the statistical analysis been performed appropriately and rigorously? 

Reviewer #1: Yes

Reviewer #2: Yes

3. Have the authors made all data underlying the findings in their manuscript fully available?

Reviewer #1: No

Reviewer #2: No

4. Is the manuscript presented in an intelligible fashion and written in standard English?

Reviewer #1: Yes

Reviewer #2: No

5. Review Comments to the Author

Reviewer #1: Thank you very much for sending in this review as well as thank the author for the great work.

They set out to determine the prevalence of alcohol consumption in ART naive patients initiating HIV care and assess whether alcohol consumption was associated with time to ART initiation, mortality and patient retention in care in East Africa.

They found that alcohol consumption was associated with lower rates of ART initiation and lower patient retention in care.

The study used data from clinics under the IeDEA consortium with clients enrolled between 2013 and 2014.

What is not clear is why data of a rather old cohort was used and more so the duration of followup is not stated and therefore it is very difficult to ascertain what the advantage of the old cohort was given that the authors do not state this clearly.

The number of subjects enrolled in the study from multiple clinics is 765 participants. whether this number was enough to demonstrate the outcomes was not illustrated and could partly explain the lack of effect of hazardous and hyper drinkers. The authors could help the readers by describing the power the study.

The authors should demonstrate what new knowledge this paper is adding to the body of knowledge and most importantly how to generalize and apply these results in the current test and treat era, where viral loads are available for treatment monitoring and where programs have intensified patient retention activities.

Introduction: Line 22.... The authors quote reference 11 about causality, what is not clear is the applicability of this reference given that the authors are also not inferring any causality.

The study used ART naive patients initiating care although many countries have reached a bigger proportion of their PLHIV and initiated them on ART with the current policy of teats and treat , what is not clear is how this study can be applied in the setting of test and treat. This weakness should be thoroughly discussed in the discussion section including the failure by the authors from obtaining Viral load data should be further discussed.

Methods:Predictors used in the analysis-- The authors should check the definition of Non drinkers and drinkers.

conduct a power analysis or sample size calculation.

The authors did not mention whether only 765 participants were the clients who came for care in the study clinics and indeed enrolled in care in those multiple clinics during the study enrollment period. How many were not enrolled? what were the other reasons for exclusion if any ?

Reviewer #2: See attached word document for reviewer comments to the author which address the above questions and provide feedback on the submission.

6. PLOS authors have the option to publish the peer review history of their article (what does this mean?). If published, this will include your full peer review and any attached files.

Reviewer #1: No

Reviewer #2: Yes: Sarah B. Puryear

---

## [Author Response · Author response to Decision Letter 0]

26 Jun 2020

Reviewer # 1

What is not clear is why data of a rather old cohort was used and more so the duration of follow-up is not stated and therefore it is very difficult to ascertain what the advantage of the old cohort was given that the authors do not state this clearly.

We state (see below) that some of the outcomes, particularly mortality, were rather rare in this population, so the power to detect any differences was limited.

The number of subjects enrolled in the study from multiple clinics is 765 participants. Whether this number was enough to demonstrate the outcomes was not illustrated and could partly explain the lack of effect of hazardous and hyper drinkers. The authors could help the readers by describing the power the study.

We state the advantages of using this particular cohort, which have to do with the fact that this is a research study (versus a study from routinely collected data), and we also outline the limitations of using a pre-UTT cohort in the limitations of the study (see below).

Duration of follow-up is not stated

The initial plan was to follow-up subjects for six months after enrollment into the study. The three participating sites had staggered start times based on receipt of regulatory approval. All patients had the same completion date, with the longest follow-up period for a subject being 433 days. 

The authors could help the readers by describing the power the study. Conduct a power analysis or sample size calculation.

Power studies and sample-size calculations are helpful at the stage of study design. They are not relevant to be used in retrospect. It is obvious, and was discussed at length in the text, that, due to very few events in some of the analyses (particularly with respect to mortality), the study had inadequate power to deect all but the most extreme differences. 

Generalize and apply these results in the current test and treat era, where viral loads are available. …the failure by the authors from obtaining Viral load data should be further discussed

An attempt was made to use viral load in this study but there were insufficient data to support an analysis (just 29 entries). As pointed out in the discussion, the most significant finding is that, from a programmatic point of view, drinkers seem to have a significant lower propensity (hazard) to initiate ART. Whether this is the case with the implementation of new guidelines remains to be seen, probably by conducting a new study utilizing a new cohort. In other words, it remains to be seen whether, in the more simplified environment of ART initiation during the universal-test-and-treat era, the impact of factors which delay treatment initiation persists. Certainly, the impact of other factors (e.g., adherence counseling, availability of laboratory tests at the start of therapy, etc.), which delayed ART initiation even among otherwise eligible patients, has receded in recent months, after adoption of UTT by virtually every country in the world (see Tymejczyk et al., J Inf Dis 2019). We have added this comment in the Discussion.

Introduction: Line 22.... The authors quote reference 11 about causality, what is not clear is the applicability of this reference given that the authors are also not inferring any causality.

Actually, reference 11 (Monroe et al.., JAIDS, 2016) speaks about an association and not causation, so it is similar to our study. In addition, the study of Monroe and colleagues does not address issues of delay of ART initiation and did not ascertain what happened to patients who were lost to the program, which, depending on the reason (unreported death versus transfer versus frank disengagement from care) would lead to possibly different conclusions. Our study, in addition to directly addressing the question of retention, it ensured that patients who were no longer retained at our sites were or were not alive and in care, by tracing those who stopped attending clinic in the community.

The study used ART naive patients initiating care although many countries have reached a bigger proportion of their PLHIV and initiated them on ART with the current policy of teats and treat, what is not clear is how this study can be applied in the setting of test and treat. This weakness should be thoroughly discussed in the discussion section including the failure by the authors from obtaining Viral load data should be further discussed.

We mention this issue in several points throughout the manuscript (see also below). The short answer is that this study cannot address what the nexus between alcohol and ART initiation and retention will be in the era of universal test and treat (UTT). However, this study is useful as further evidence that ignoring alcohol use (and, for that matter, other factors which may affect retention) in the rush to initiate everyone on therapy, may be counterproductive in the long run.

Predictors used in the analysis.

We have added that patient-level characteristics (demographic factors like age and gender, plus disease-related factors like CD4 count, World Health Organization (WHO) HIV disease stage), logistical issues (e.g., distance from clinic) and clinic-level characteristics, such as location of clinic and level of care, were collected by the study and were used in the analyses. 

The authors should check the definition of Non drinkers and drinkers.

This is a good point. In our manuscript “non-drinkers (AUDIT > 0) versus drinkers (AUDIT = 0);” -it should be the other way around. We have corrected this in the paper.

 

The authors did not mention whether only 765 participants were the clients who came for care in the study clinics and indeed enrolled in care in those multiple clinics during the study enrollment period. How many were not enrolled? What were the other reasons for exclusion if any?

The following is an excerpt from the screening log from two of our sites in Kisumu, Kenya. It appears, reassuringly enough, that the majority of exclusions concerned protocol violations, with only a small number of patients who were ultimately not enrolled having done so because of refusal to participate in the study.

Site No Screened No Enrolled Not Enrolled- 63

Kisumu 232 169 Reason Number

 Transfer In- 33

 Declined Enrollment 12

 Known Positives 5

 Under the age of 18 12

 Very sick with T.B- 1

Suba Site 89 76 Declined Enrollment 5

 Under the age of 18 8

Totals 

Screened 321 

Enrolled 245 

Not Enrolled 76 

Reasons 

Transfer In 33 

Declined Enrollment 17 

Known Positives 5 

Under the age 20 

Very sick clients 1 

Our site in Uganda reported the following:

Number of patients screened: 490

Number of patients enrolled: 264

The declines had the following issues:

• Not interested

• Declined to give a reason

• Too weak

• Below Age

• Needed approval from partner

• Stigma issues

• Mental issues

• Time bad

• House maid

• Prisoner

Unfortunately, we have not yet been able to ascertain how many of those who declined participation (226/490) were in each of these groups and obviously, some of the issues listed may be associated with alcohol use (e.g., “mental issues”) or even some among those who were “not interested” or “declined to give a reason”, or stated “time [was] bad” may have had their decision to participate influenced by alcohol consumption leading to some bias in the outcome. Nevertheless, to the extent that the experience in Uganda is similar as to that in Kenya, the majority of exclusions were related to logistical issues or protocol violations. 

Reviewer # 2

TITLE

The title as written is vague. Suggest clarifying the intended meaning of “impact” in the title, e.g. “Prevalence of Alcohol Use and Impact on HIV Care Outcomes in ART-naïve Adults Enrolling in HIV Care in East Africa”

We have modified the title verbatim.

ABSTRACT

Methods: Would be more specific than “East Africa” as this affects interpretation of alcohol use prevalence; recommend specifying Kenya & Uganda

We have limited geographic references to Kenya and Uganda rather than the entire region of East Africa as suggested.

Methods: Because primary outcomes are measured over time, please include the duration of follow-up

As above, the maximum duration of follow-up was 433 days

Methods: Include the comparison groups for each classification, i.e. non-drinkers for all? Or non-hazardous, non-hyper, etc.?

• Non-drinkers vs drinkers

• Non-hazardous drinkers (which includes non-drinkers) vs hazardous drinkers

• Non-hyper drinkers (which includes non-drinkers and hazardous drinkers) vs. hyper drinkers

Results: Suggest clarifying “any alcohol use was associated..” as hazardous and hyper use were not found to be associated with retention or ART outcomes.

We agree. This was done 

Results: Given mortality is a primary outcome of the study, results regarding the association should be included within the abstract

We agree. We have included in the abstract a mention that mortality differences were not observed.

Conclusions: The phrasing “significant delays in ART initiation” implies a period, however this is not provided in the results. I.e. is a significant delay 1 week or 1 year?

The statistical analysis modeled a quantity related to the hazard (or propensity) of initiating ART. This is called the “sub-distribution hazard” (Fine & Gray models) of initiating ART between drinkers and non-drinkers subject to the (competing) risk of death prior to initiating therapy. Since this sub-distribution hazard of initiating ART was significantly lower among people living with HIV who consumed alcohol, the cumulative incidence of initiating treatment (i.e., the probability of starting therapy before a given time) was lower for these individuals. We attempted to demystify this concept for a non-statistical audience by referring to this as a “delay in ART initiation”. We are now referring to “a lower probability of initiating therapy” rather than delays in therapy initiation.

INTRODUCTION

2nd paragraph: “impact on initiation of ART and retention in care yet to be thoroughly examined” is perhaps an oversimplification of the literature. Certainly, there is a gap, but a more nuanced discussion of the data would help to contextualize these findings. 

We have pointed out the fact that the impact of alcohol consumption on initiation of ART and retention in care have not been fully examined and stated that this is one in a series of studies on the subject.

Mention of "few EBM interventions” distracts from the point of the paragraph; suggest deleting

Done

In general, the points raised here don’t connect to the study being done: i.e. the study does not introduce new measure of alcohol exposure, new measurement of retention, or adjust for more confounders than past studies

The “novelty” of this study is to assess what the possible impact of alcohol consumption is on the speed of ART initiation and retention in care. We clarify this in the introduction.

METHODS

In general, the ordering of this section is confusing. Recommend that the opening sentence include not only the study design type, but also the study period, study purpose and setting. I.e. “From January 2013 to June 2014, we conducted a prospective observational study to determine X among adults entering HIV care in Uganda and Kenya.” This will clarify and contextualize the subsequent detailed methods section. Furthermore, the subheading “Study Population” should be divided to include a section on “Measures.” Several portions of the text included under “Statistical methods” are descriptions of measures, rather than statistical methods and should be re-organized appropriately. 

We have significantly reorganized this section per the reviewer’s comments.

Study Design/Ethics: Was referral for alcohol counseling or treatment offered to hazardous and hyper drinkers?

Yes, all patients were given a brochure with information on HIV and alcohol use regardless of their responses to the AUDIT questionnaire. This brochure had information on how to contact a health provider for assistance or to ask more questions. This was added in the methods.

Study setting and SOC: The study period is mentioned but not defined until later in the manuscript; please add study years with respect to ART eligibility criteria to aid interpretation of the SOC being described. Both Kenya and Uganda updated ART guidelines to expand treatment to all individuals with a CD4<500 in 2014. Was this implemented during the study time period? Were all individuals started on ART based on a cut off of 350 (i.e. were pregnant women or those with a high CD4 and OIs excluded?)

Although the CD4 cutoff was 350 cells/μl at the time of study initiation, the sites were already moving to starting patients on ART who had CD4 counts <500 cells/μl. Women who were pregnant and those with opportunistic infections were also eligible for ART at CD4 counts >500 cells/μl. This was also added to the methods.

Study setting and SOC: Please clarify ART eligibility criteria further (1) how were pregnant women and (2) those with OIs treated if CD4 > threshold? 

See above

Study population: Recommend a sub-heading of “Measures” after describing and participants and before describing AUDIT

Sub-headings have been added as part of the overall reorganization of this section.

Study population: Please provide reference for AUDIT validation in sub-Saharan African setting

The following reference has been added in this section:

Chishinga N, Kinyanda E, Weiss HA et al. Validation of brief screening tools for depressive and alcohol use disorders among TB and HIV patients in primary care in Zambia. BMC Psychiatry, 11, 75 (2011). https://doi.org/10.1186/1471-244X-11-75

Study population: You reference the “AUDIT version utilized”: was this not the standard AUDIT tool? If non-standard, please include in the supplemental information. The standardization of procedures used to assess alcohol use are unclear. If additional or supplemental questions were asked and conflicted with answers to the standard 10-question screen, which answer was used in the analysis? While standard to include a chart of types of alcohol and volume to aid classification of alcohol use, additional questions about frequency are not typically standard. 

We have indicated that the standard AUDIT test was used and have removed the word “version”, which in retrospect was unfortunate and caused confusion. We also state that a number of visual aids were used to attempt to ascertain the alcohol content of home brews so that they could be entered in the standardized AUDIT tool.

Statistical methods: The definition of “retention in care” is unclear. Firstly, “retention in care,” as defined, seems to measure “retention in care failure” as it is “patients actually disengaged from care”. Second, earlier in the manuscript, it is stated that a normal interval between appointments is up to 6 months; however, retention in care failure is being assessed at 2 months following the last (non-missed?) appointment.

Because follow-up schedules change from time to time, we have defined as a missed visit, leading to a designation as “lost to program” any visit that does not occur within two months from the next scheduled visit. This takes care of varying frequencies in appointments. We have clarified in this in the text. We have defined as “retention in care” from a patient’s rather a program’s perspective as the complement event of a true disengagement from care (or otherwise known as a “gap in care”). So a patient is retained if they are in care; anywhere. Given that this study, unlike most routine protocols, included aggressive patient outreach to establish both patients’ vital status and their access to care, after failing to keep a clinic visit at one of our program facilities, we were able to assess true disengagement from care and, conversely, true retention in care. We have clarified this in the text.

Statistical methods: The sentence “IN this manner, we were able to estimate” is editorial and does not describe methodology. Suggest deleting. 

I think that this phrase actually clarifies that it is the retention in care from the perspective of the patient that we try to measure. This is an important feature of this study, in contrast to most similar published research studies. Being able to estimate retention in this manner removes a major limitation present in the vast majority of studies, where retention in care means continuous care at the index program. We have added more forceful language about the fact that it is retention in care from the patient’s perspective that this study was capable of estimating.

Statistical methods: For those who were consistently retained in care (i.e. did not require tracing), how was censoring done? 

Out of study subjects who were consistently retained in care, those who at the end of the follow up period did not experience any event were censored. 

Statistical methods: How was the mortality outcome ascertained? Were death registers utilized?

Mortality was ascertained in multiple ways. “Passively”, or through reports to the clinic by family members or healthcare providers, or “actively”, by reaching out to the patient’s family, community or village elders, either by telephone or by sending a community worker to the patient’s disclosed location to determine their vital status. 

Statistical methods: Please provide a clear definition of the 3rd primary outcome, time from enrollment to ART initiation. This is later referred to as delays in ART initiation: please clearly define the outcome and use consistent terminology. Is ART initiation restricted based on ART eligibility by CD4 measured at the relevant visit?

Please see above. We have refrained from referring to lower incidence in ART initiation as a “delay”, particularly since the data are moot on this issue. Our use of the term “delay of ART initiation” emanates from the fact that, during this period of time, patients could meet eligibility criteria to start ART, but the actual decision to start was based by the clinician’s or the patient’s assessment of “ART readiness”, i.e., the patient’s ability to take ART medications on a daily basis. Patients may have had extenuating circumstances (such as alcohol use) that could hinder their ART readiness which resulted in them delaying ART initiation until barriers to being fully compliant to the drugs before starting were eliminated. This manifests in the data as a longer time until ART initiation (and, by extension, a lower incidence of ART start in these patients). Nevertheless, there is no data value that explicitly encodes a decision to defer ART initiation, resulting in the less explicit reference to this as a decreased incidence of initiation of ART. 

Statistical methods: For all primary outcomes, what was the timeline from enrollment to outcome measurement? This is critical for interpretation of the outcomes; retention in care over 3 months is vastly different than over 2 years, for example.

The maximum length of follow-up was 433 days.

Predictors used in the analyses: For the alcohol use categories, were non-drinkers (AUDIT=0) included in the groups non-hazardous drinkers? In non-hyper drinkers?

Yes. Categories were dichotomized according to level of drinking. So non-hazardous drinkers and non-hyper drinkers would include non-alcohol users. We suspect that this may have been the reason for the equivocal results observed in these groups, as complete abstention from alcohol appeared to be strongly related to earlier ART initiation (or higher cumulative incidence of treatment initiation after enrollment). Having sequestered the non-alcohol use category (in, for example, a three-way analysis of no alcohol, non-hyper drinking (among alcohol users). 

Predictors used in the analyses. Regarding the terminology “hyper-drinkers”, is this used elsewhere in the literature? If yes, please provide a reference. If not, what was the rationale for combining standard categories, given mention in the introduction that non-standard definitions of alcohol use limit comparison across studies?

This is our choice of terminology, in an attempt to differentiate hazardous drinking from excessive drinking (without using socially or morally fraught terminology). These categories correspond to standard Audit score cutoffs for the highest level of alcohol consumption.

RESULTS

Participant characteristics: How many patients were screened and excluded based on the criteria outlined in methods? Distribution of reasons for exclusion?

See above.

What percent met eligibility for ART at enrollment? At subsequent visits? What was the mean follow up time and number of visits for participants?

A significant proportion of subjects was eligible at initiation (263 out of 765 or 34.4%). The median number of visits was 6, with a range of 1 to 17.

“Initially 25% lost”: When is initially? Is this the follow up interval?

This refers to the number of study participants who were lost to program. These subjects were thus lost from a programmatic perspective. In a routine analysis, like those in the majority of similar publications in the literature, this would effectively be the proportion of patients not retained in care (again, from the perspective of the program). However, as we delineate in the manuscript, this estimate would significantly underestimate retention in care (and, by extension, risk factors for non-retention) as this group includes people who were truly disengaged from care, others who were deceased and still others who were in care elsewhere (i.e., had an undocumented or “silent” transfer to another facility).

 

DISCUSSION

In general, the null findings for hazardous and hyper-drinkers seem most like attributable to a small sample size with a rare outcome that is unable to detect significance. The discussion should include attention to this possibility.

We have addressed this as part of the limitations of the study. Another possibility is that, once one gets away from the alcohol use/no alcohol use dichotomy, the comparison groups may become more variable as well.

A more robust discussion of the studies limitations, including the assumptions of the statistical model, small sample size, observational nature/incomplete data, etc., is merited. 

We have attempted to make a fuller accounting of the limitations of the study in the Discussion. Please see also previous comment.

CONCLUSION

The conclusion is over-stated. The study did not find that all amounts of alcohol use were association with lower rates of ART initiation and lower retention in care.

With respect, we don’t want to get too far (become to equivocal) with respect to the conclusion that alcohol use (including higher levels of use) was associated with challenges in ART initiation and retention. As I point out earlier in this response to comments, low power and heterogeneous comparison groups may have been the “culprits” for the negative results with respect to higher levels of drinking. In addition, given that the study was based on self-reported alcohol use levels (another limitation explicitly stated in the Discussion), there may also be greater heterogeneity in what constitutes “high” levels of alcohol use, to a much greater extent than someone stating whether they consume alcohol or not.

The 3rd component of 90-90-90 is viral suppression, not retention in care; furthermore, retention in care is an intermediary step preceding suppression, but does not serve as a proxy for viral suppression

Noted. We have removed references to retention in care as the third pillar of the 90-90-90 target.

The conclusion that “its evaluation appears to be a significant target for intervention is overstated; however fair to conclude that screening for alcohol use in ART naïve patients enrolling in care may be able to identify patients at higher risk for delaying ART start despite eligibility and not being retained in care. 

We have revised this sentence per reviewer recommendation.

Table 2: 

Change in the ordering of columns adds confusion. Please move binary drinkers to the first column to compare with Tables 1 & 3. 

Done

---

## [Decision Letter · Decision Letter 1]

13 Aug 2020

PONE-D-19-24950R1

Lower rates of ART initiation and decreased retention among ART-naïve patients who consume alcohol enrolling in HIV care and treatment programs in Kenya and Uganda

PLOS ONE

Dear Dr. Yiannoutsos,

Thank you for submitting your manuscript to PLOS ONE. After careful consideration, we feel that it has merit but does not fully meet PLOS ONE’s publication criteria as it currently stands. Therefore, we invite you to submit a revised version of the manuscript that addresses the points raised during the review process.

We look forward to receiving your revised manuscript.

Kind regards,

Joel Msafiri Francis, MD, MS, PhD

Academic Editor

PLOS ONE

Reviewers' comments:

Reviewer's Responses to Questions

**Comments to the Author**

1. If the authors have adequately addressed your comments raised in a previous round of review and you feel that this manuscript is now acceptable for publication, you may indicate that here to bypass the “Comments to the Author” section, enter your conflict of interest statement in the “Confidential to Editor” section, and submit your "Accept" recommendation.

Reviewer #1: (No Response)

2. Is the manuscript technically sound, and do the data support the conclusions?

Reviewer #1: Yes

3. Has the statistical analysis been performed appropriately and rigorously? 

Reviewer #1: Yes

4. Have the authors made all data underlying the findings in their manuscript fully available?

Reviewer #1: No

5. Is the manuscript presented in an intelligible fashion and written in standard English?

Reviewer #1: Yes

6. Review Comments to the Author

Reviewer #1: The authors have commendably addressed most of the comments highlighted at the previous review.

There are no line numbers to facilitate quick review however below are some suggestions to tighten the message and make it easier for the audience to understand.

Methods :

1. Study setting and standard of care

Paragraph two: ......Individuals enrolling in care have a CD4 count drawn ...... : could this be rephrased to read "Individuals enrolling in care have blood drawn for CD4 cell count "?

2. Study procedures

a) The Alcohol Use Disorder Screening Test (AUDIT) ......> Change word "screening" to "Identification"

b)Please provide the period of recall for alcohol use in this section - 1 yr?

3. Statistical analyses

Paragraph two ......different dichotomous level of alcohol consumption (i.e., binary, hazardous and hyper drinking, the exposure)..... "Binary " changed to word "drinkers" or "any drinking"

Results

Competing risk analysis

4....... Given the low rates of non-retention (10.6%, Table 2), this is approximately equal 76.6% higher odds or risk of non-retention.....

a) Improve sentence to read 77%( similar visually to the aSHR 1.77) instead 76.6%

b) Review the use of the words " higher odds or risk". The author's reference 26 [Practical recommendations for reporting Fine‐Gray model analyses for competing risk data] recommends that the SHR is best reported as a rate rather that risk since the aSHR does not in and of itself quantify the magnitude of the effect of "any drinking" on the Cumulative incidence function of non retention.

Discussion

5) sentence 1 . This study found a strong association between alcohol consumption and non-retention in

HIV care. - Suggestion to rephrase as "This study found a strong association between alcohol consumption within the last *** insert recall period) and non-retention in HIV care.

6) Consequently, with less than 4% of study subjects dying during the study, there was very small power to detect any potential effects of alcohol consumption on mortality. Remove wording very small power and could possibly rephrase to " Consequently, with less than 4% of

study subjects dying, our study was not powered to detect any potential effects of alcohol consumption on mortality.

7) Rephrase sentence " In addition, dichotomizing the

data inCombined, these limitations in our data collection may account for finding an association

of alcohol consumption with decreased retention in care and delays in ART initiation, but not

confirming the same findings among those at the highest alcohol consumption classifications."

7. PLOS authors have the option to publish the peer review history of their article (what does this mean?). If published, this will include your full peer review and any attached files.

Reviewer #1: No

---

## [Author Response · Author response to Decision Letter 1]

28 Aug 2020

Reviewer # 2

METHODS

Study setting and standard of care

Paragraph two: ......Individuals enrolling in care have a CD4 count drawn ...... : could this be rephrased to read "Individuals enrolling in care have blood drawn for CD4 cell count "? 

Done

Study procedures

a) The Alcohol Use Disorder Screening Test (AUDIT) ......> Change word "screening" to "Identification"

Done

b) Please provide the period of recall for alcohol use in this section - 1 yr?

The term “over the past one year” was added to the sentence “The Alcohol Use Disorder Identification Test (AUDIT) questionnaire, which has been validated in a number of primary care medical settings, was used by a trained research assistant to collect patient’s alcohol use data over the past one year.”

Statistical analyses

Paragraph two ......different dichotomous level of alcohol consumption (i.e., binary, hazardous and hyper drinking, the exposure)..... "Binary " changed to word "drinkers" or "any drinking"

We have removed all accurate but, admittedly, awkward terms like “binary” or “dichotomous” drinking in favor of more colloquial terminology like “any alcohol use”. For example, the above paragraph was changed to “We performed three separate analyses, each investigating the possible effect of different levels of alcohol consumption (i.e., any alcohol use, hazardous and hyper drinking, the exposure) on the likelihood (sub-distribution hazard) of each of the three events of interest (ART initiation, death, and non-retention in care).” (changes bolded).

Results

Competing risk analysis

Given the low rates of non-retention (10.6%, Table 2), this is approximately equal 76.6% higher odds or risk of non-retention ... [i]mprove sentence to read 77%( similar visually to the aSHR 1.77) instead 76.6%

Done

Review the use of the words " higher odds or risk". The author's reference 26 [Practical recommendations for reporting Fine‐Gray model analyses for competing risk data] recommends that the SHR is best reported as a rate rather that risk since the aSHR does not in and of itself quantify the magnitude of the effect of "any drinking" on the Cumulative incidence function of non retention.

Done. For example, the above sentence now reads “Given the low rates of non-retention (10.6%, Table 2), this is approximately equal 77% higher rate of non-retention. Other alcohol exposure classifications (i.e., hazardous versus non-hazardous…” (change bolded). 

Discussion

Sentence 1 . This study found a strong association between alcohol consumption and non-retention in

HIV care. - Suggestion to rephrase as "This study found a strong association between alcohol consumption within the last *** insert recall period) and non-retention in HIV care.

Done with a reference to the one-year recall applied to the one-year recall

Consequently, with less than 4% of study subjects dying during the study, there was very small power to detect any potential effects of alcohol consumption on mortality. Remove wording very small power and could possibly rephrase to " Consequently, with less than 4% of study subjects dying, our study was not powered to detect any potential effects of alcohol consumption on mortality.

Done

Rephrase sentence " In addition, dichotomizing the data in

Combined, these limitations in our data collection may account for finding an association

of alcohol consumption with decreased retention in care and delays in ART initiation, but not

confirming the same findings among those at the highest alcohol consumption classifications."

Done

---

## [Decision Letter · Decision Letter 2]

30 Sep 2020

PONE-D-19-24950R2

Lower rates of ART initiation and decreased retention among ART-naïve patients who consume alcohol enrolling in HIV care and treatment programs in Kenya and Uganda

PLOS ONE

Dear Dr. Yiannoutsos,

Thank you for submitting your manuscript to PLOS ONE. After careful consideration, we feel that it has merit but does not fully meet PLOS ONE’s publication criteria as it currently stands. Therefore, we invite you to submit a revised version of the manuscript that addresses the points raised during the review process.

We look forward to receiving your revised manuscript.

Kind regards,

Joel Msafiri Francis, MD, MS, PhD

Academic Editor

PLOS ONE

Reviewers' comments:

Reviewer's Responses to Questions

**Comments to the Author**

1. If the authors have adequately addressed your comments raised in a previous round of review and you feel that this manuscript is now acceptable for publication, you may indicate that here to bypass the “Comments to the Author” section, enter your conflict of interest statement in the “Confidential to Editor” section, and submit your "Accept" recommendation.

Reviewer #1: All comments have been addressed

Reviewer #3: (No Response)

2. Is the manuscript technically sound, and do the data support the conclusions?

Reviewer #1: Yes

Reviewer #3: Yes

3. Has the statistical analysis been performed appropriately and rigorously? 

Reviewer #1: Yes

Reviewer #3: Yes

4. Have the authors made all data underlying the findings in their manuscript fully available?

Reviewer #1: No

Reviewer #3: No

5. Is the manuscript presented in an intelligible fashion and written in standard English?

Reviewer #1: Yes

Reviewer #3: Yes

6. Review Comments to the Author

Reviewer #1: (No Response)

Reviewer #3: This manuscript is well written and clear. The data analyses sufficiently support the conclusions.

I recommend minor edits, specifically to terminology used throughout the manuscript to describe persons with HIV and persons who drink.

Consistent with current literature, the authors should consider using 'persons with HIV (PWH)' in place of 'people living with HIV (PLWH)'.

The authors should also reconsider their descriptions of persons who consume alcohol as well as their classification of levels of alcohol use/drinking. Consistent with current literature, the use of the word 'drinkers' throughout the manuscript should be changed to 'persons who consume/drink alcohol', or 'persons who engage in alcohol consumption', etc. Along the same lines, levels of alcohol use may be described as 'heavy alcohol use or drinking' and 'very heavy alcohol use or drinking' to match the corresponding levels described in the manuscript as 'hazardous' and 'hyper'. 'Harmful drinking' and 'alcohol dependent/dependency' are terms that are not in line with current terminology, which is: 'at risk for alcohol use disorders'. The use of the term 'hyper' in particular is not standard. The use of the term 'non-drinkers' should be reconsidered and perhaps replaced with 'abstainers' instead.

7. PLOS authors have the option to publish the peer review history of their article (what does this mean?). If published, this will include your full peer review and any attached files.

Reviewer #1: No

Reviewer #3: **Yes: **Nneka I. Emenyonu

---

## [Author Response · Author response to Decision Letter 2]

30 Sep 2020

We have revised the terminology changing virtually all references to "drinking" as "alcohol consumption", "hazardous drinking" as "engagement in hazardous alcohol consumption", "hyper drinking" as "harmful alcohol consumption" (per AUDIT designation) wherever possible. We did not change a reference to "heavy drinkers" as this was the designation from another study. We have changed PLWHIV to PWH for people with HIV as requested by the reviewer.

---

## [Editor Report · Decision Letter 3]

1 Oct 2020

Lower rates of ART initiation and decreased retention among ART-naïve patients who consume alcohol enrolling in HIV care and treatment programs in Kenya and Uganda

PONE-D-19-24950R3

Dear Dr. Yiannoutsos,

We’re pleased to inform you that your manuscript has been judged scientifically suitable for publication and will be formally accepted for publication once it meets all outstanding technical requirements.

Kind regards,

Joel Msafiri Francis, MD, MS, PhD

Academic Editor

PLOS ONE
---

## [Editor Report · Acceptance letter]

16 Oct 2020

PONE-D-19-24950R3 

Lower rates of ART initiation and decreased retention among ART-naïve patients who consume alcohol enrolling in HIV care and treatment programs in Kenya and Uganda 

Dear Dr. Yiannoutsos:

I'm pleased to inform you that your manuscript has been deemed suitable for publication in PLOS ONE. Congratulations! Your manuscript is now with our production department. 

Kind regards, 

on behalf of

Dr. Joel Msafiri Francis 

Academic Editor

PLOS ONE